# Syntactic Substitutability as Unsupervised Dependency Syntax

**Jasper Jian** [1] and **Siva Reddy** [2,3]

[1] Stanford University
[2] Mila – Quebec AI Institute and McGill University
[3] Facebook CIFAR AI Chair
jjian@stanford.edu, siva.reddy@mila.quebec

## Abstract

Syntax is a latent hierarchical structure which underpins the robust and compositional nature of human language. In this work, we explore the hypothesis that syntactic dependencies can be represented in language model attention distributions and propose a new method to induce these structures theory-agnostically. Instead of modeling syntactic relations as defined by annotation schemata, we model a more general property implicit in the definition of dependency relations, *syntactic substitutability*. This property captures the fact that words at either end of a dependency can be substituted with words from the same category. Substitutions can be used to generate a set of syntactically invariant sentences whose representations are then used for parsing. We show that increasing the number of substitutions used improves parsing accuracy on natural data. On long-distance subject-verb agreement constructions, our method achieves 79.5% recall compared to 8.9% using a previous method. Our method also provides improvements when transferred to a different parsing setup, demonstrating that it generalizes.

## 1  Introduction

In recent years, large pretrained language models (LLMs), like BERT (Devlin et al., 2019), have led to impressive performance gains across many natural language processing tasks. This has led to a line of work attempting to explain how natural language understanding might occur within these models and what sorts of linguistic properties are captured. Going one step further, we explore the hypothesis that syntactic dependencies can be *extracted* from LLMs without additionally learned parameters or supervision.

Previous work on syntax has tested (1) whether language models exhibit syntactically dependent

---

Work done while the first author was at McGill University. The code and data can be found at https://github.com/McGill-NLP/syntactic-substitutability.

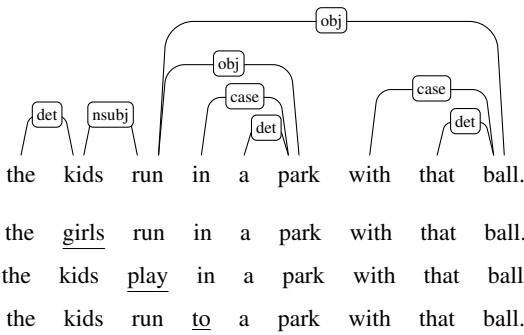

Figure 1: Syntactic relations represent intersubstitutability: the nominal subject 'kids' can be substituted with another noun 'girls' without affecting syntactic well-formedness. Swapping the verb 'run' with 'play' and the preposition 'in' with 'to' are other examples for this sentence. Substitutions define a set of sentences that can be used to model this property during parsing.

*behaviour* like long-distance subject-verb agreement (Marvin and Linzen, 2018; Gulordava et al., 2018; Goldberg, 2019), and (2) whether syntactic *structures* are retrievable from model-internal representations or mechanisms (Hewitt and Manning, 2019; Htut et al., 2019; Limisiewicz et al., 2020). While the former approach is theory-agnostic, as it does not require a specific syntactic form to be defined, it lacks the interpretability that inducing explicit structures provides, as we do here.

Instantiating the latter approach, Hewitt and Manning (2019) train a probe in order to project model representations of words into a new vector space where a maximum spanning tree algorithm (MST) can be applied to induce the desired syntactic parse. However, it is not clear whether such a method relies solely on the information already present in the model, or whether the trained probe is contributing model-external knowledge (Belinkov, 2022). A less ambiguous approach is instead to use model-internal distributions directly as input to the tree induction algorithm without additional training. In this vein, previous work has made use

of attention distributions from transformer-based LMs (e.g. Raganato and Tiedemann, 2018; Htut et al., 2019). This parallels observations made by Clark et al. (2019) that certain attention heads in BERT correspond to dependency relations. However, given the large amount of information present in LLMs, nothing constrains the extracted parses to be *syntactic* parses when representations are used directly.

In this paper, we propose a novel method to distill syntactic information by modelling a general property of syntactic relations which is independent of any specific formalism. This property, *syntactic substitutability*, captures the intuition that syntactic structures define categories of intersubstitutable words – illustrated in Figure 1. We make use of this notion by enumerating syntactically invariant sentences which can then be exploited together to induce their shared syntactic structure. Our primary goal is to investigate whether modeling syntactic substitutability better extracts syntactic information from attention mechanisms, resulting in more accurately induced parses. Inducing structures without relying on specific annotation schemata also allows us to better understand how the syntactic relations represented in a model might be similar to existing theoretical proposals.

We demonstrate that our method, Syntactic Substitutability as Unsupervised Dependency Syntax (**SSUD**) leads to improvements in dependency parsing accuracy. As more substitutions are used, parsing scores increase. We also quantitatively show that the induced parses align more with an annotation schema where function words are treated as heads (Experiment 1). When tested on long-distance subject-verb agreement constructions, SSUD achieves an increase in recall of >70% compared to a previous unsupervised parsing method (Experiment 2). We also demonstrate how our method can be transferred to and improve different parsing algorithms, showing that SSUD generalizes effectively (Experiment 3).

## 2 Related work

Our work is related to a long tradition of unsupervised syntactic parsing, for example, the generative DMV model (Klein and Manning, 2004) and the compound Probabilistic Context Free Grammar model (Kim et al., 2019) for constituency parsing. Motivations for this stem from the idea that syntactic parses can be induced by computing an MST over scores between words in a sentence that represent how likely it is for two words to be in a syntactic dependency or how likely a span corresponds to a syntactic constituent.

We seek to work with scores directly derivable from LLMs, following previous proposals which have used attention distributions of transformer-based models (Vaswani et al., 2017) to calculate scores. Examples in dependency parsing include Raganato and Tiedemann (2018) who use neural machine translation models, and Htut et al. (2019) and Limisiewicz et al. (2020) who use BERT. For constituency parsing, Kim et al. (2020) propose a method based on the syntactic similarity of word pairs, calculated as a function of their attention distributions. The use of attention distributions for parsing tasks is supported by the observation made by Clark et al. (2019) that certain attention heads correspond to syntactic dependencies in BERT. However, they also observe that attention heads do not only capture syntactic information, but also other relationships like coreference. Our method proposes syntactic substitutability to address this issue, as motivated in the next section. Previous work from Limisiewicz et al. (2020) proposes an algorithm to select syntactically relevant heads, but we contend that this is maximally effective if the distributions within a single head also *only* capture syntactic information. We return this idea in Experiment 3 and investigate the effect of adding SSUD to this algorithm.

Other complementary methods use BERT's contextualized representations to perform parsing. For example, Wu et al. (2020) propose a method which calculates scores based on the 'impact' that masking a word in the input has on the representations of the other words in the sentence, and Hewitt and Manning (2019) train a probe with supervision to project vectors into a 'syntactic' space. Another approach is using BERT's masked language modeling objective to compute scores for syntactic parsing. Work in this vein include Hoover et al. (2021) and Zhang and Hashimoto (2021), motivated by a hypothesis stemming from Futrell et al. (2019) that syntactic dependencies correspond to a statistical measure of mutual information.

Lastly, while the non-parametric use of substitutability for syntactic parsing has not been previously proposed, parallels can be drawn to work in language model interpretability. Papadimitriou et al. (2022) show that BERT systematically learns

to use word-order information to syntactically distinguish subjects and objects even when the respective nouns are swapped. This is a special case of substitution: the syntactic structure of these sentences is invariant despite their non-prototypical meaning. We take these results to mean that BERT has a knowledge of syntax that is robust to substitution, and as a result, substitutability may be an effective constraint.

# 3 Syntactic Substitutability and Dependency Relations

In this section, we propose a method to model the formalism-neutral objective of substitutability within the induction of syntactic structures. This notion is often explicitly included in the definition of syntactic grammars, see for example Hunter (2021) and Mel'čuk (2009). Intuitively, intersubstitutable words form syntactic *categories* which syntactic *relations* operate on.

## 3.1 Problem statement

We wish to extract a tree-shaped syntactic dependency structure $t_s$ for a sentence $s$ from the mechanisms or representations of an LLM. We denote the target sentence $s$ of length $n$ as

$$s := <w_{(0)}, ..., w_{(i)}, ..., w_{(n-1)}>.$$

Edges in $t_s$ belong to the set of binary *syntactic* relations $R_{synt}$. The specific relations that are included are relative to specific formalisms. We define

$$Dep_{synt}(s, i, j) \in \{0, 1\}$$

which denotes whether or not two words in a sentence $s$, $w_{(i)}$ and $w_{(j)}$, are in a syntactic dependency relationship. If $Dep_{synt}(s, i, j) = 1$, then $\exists r \in R_{synt}$ s.t. $r$ relates $w_{(i)}$ and $w_{(j)}$ denoted

$$w_{(i)} \overset{r}{\leftrightarrow} w_{(j)},$$

where $w_{(i)} \overset{r}{\leftrightarrow} w_{(j)}$ denotes an *undirected* relation.

Given that relations are binary, a matrix of scores between all words in a given sentence is required before syntactic trees can be induced. In this work, we propose attention distributions of self-attention heads as candidate scores. However, any method which calculates pairwise scores between words can be used here with no change.

We denote the attention distribution for word $w_{(i)}$[1] in the given sentence $s$, of length $n$ as

---

[1] BERT's tokenizer does not tokenize on words, but rather on subwords tokens. We follow Clark et al. (2019) and sum and normalize in order to convert from attention distributions over tokens to one over words.

$$Att(s, i) := [a^s_{i0}, ..., a^s_{ii}, ..., a^s_{i(n-1)}],$$

where $a_{ij}$ refers to the attention weight from $w_{(i)}$ to $w_{(j)}$. The sentence's attention matrix, $Att(s)$, is the $n \times n$ matrix where row $i$ is equal to $Att(s, i)$.

## 3.2 Attention distributions

For each word in a sentence, attention heads in BERT compute a distribution over all words in a sentence. Each row $i \in [0, n)$ of $Att(s)$ corresponds to the attention distribution for word $w_{(i)}$.

Previous work has made use of attention distributions to extract syntactic trees by using MST algorithms over the attention matrix of a single sentence (Raganato and Tiedemann, 2018; Htut et al., 2019). The hypothesis here is that the attention scores between syntactically dependent words is higher than those that are not. Given this, the correct undirected syntactic parse can be induced, i.e. $MST(Att(s)) = t_s$, if

$$\forall (i, j) \in \{(a, b) | Dep_{synt}(s, a, b) = 1\}$$
$$\forall (y, z) \in \{(c, d) | Dep_{synt}(s, c, d) = 0\}$$
$$a^s_{ij} > a^s_{yz}. \quad (1)$$

We suggest that the assumption being made in Equation 1 is incorrect, given that attention distributions can correspond to a wide variety of phenomena – again they need not be *syntactic*. For example, an edge in the induced tree may have been predicted due to a high score resulting from coreference or lexical similarity.

## 3.3 Modeling syntactic substitutability

We propose *syntactic substitutability* as a formalism-neutral method of extracting *syntactic* information from attention distributions. Intuitively, a syntactic grammar is defined in such a way as to offer an abstraction over individual lexical items and operate on syntactic categories. Formalizing this, we make the assumption that any relation $r \in R_{synt}$ defines a set of words that can be substituted for one another in a sentence. The formal definition that we begin with is referred to as the *quasi-Kunze property* in Mel'čuk (2009). There, a relation is defined from a head word, $w_{(i)}$, to a subtree which is rooted at another word, $w_{(j)}$. For a relation to be syntactic, it must define some class of words $X$, such that subtrees which are rooted at words from $X$ can be substituted into the original sentence without affecting syntactic well-formedness. An example of this is provided in Figure 2.

the kids run in a park with the ball.
the kids run to that yard with the ball.

Figure 2: The subtree rooted at 'park' (underlined) is substituted for one rooted at 'yard.'

just thought you 'd like to know. (*Target*)
| always, simply, only | thought you 'd like to know.
just | figured, knew, think | you 'd like to know.
just thought you 'd | love, demand, have | to know.
just thought you 'd like to | help, talk, stay |.

Figure 3: A set of sentences generated via SSUD for a sentence taken from the WSJ10 dataset with example substitutions at each position listed.

We propose a modified form of this property defined in terms of the substitution of individual words since constraining substitutions to subtrees would be complex given an unsupervised process. Note, however, that this is exactly equivalent to substituting subtrees which differ only in their root word. As previously stated, we make no assumptions about the directionality of these relationships.

**Definition 1.** *Modified quasi-Kunze property*: Let $w_{(i)}$ and $w_{(j)}$ be words. For any relation $r$, if $r \in R_{synt}$, then there exists $X$, such that for any syntactic tree with relation $w_{(i)} \overset{r}{\leftrightarrow} w_{(j)}$, replacing $w_{(j)}$ with a word $x \in X$ does not affect the sentence's syntactic well-formedness.

In our framework, we assume that for any relation to be a syntactic relation, it must satisfy the *modified quasi-Kunze property* as defined above. We demonstrate that this provides a tractable objective for inducing dependency structures.

### 3.4 Generating sentences via substitution

In order to model the property in Definition 1, we generate candidate substitutions using an LLM. In this work, we use BERT itself to predict possible substitutions using masked language modeling, a task for which it was trained. We find that this generates empirically correct sentence substitutions, as in Figure 3.

We choose to substitute all open-class categories and some closed-class categories (adjectives, nouns, verbs, adverbs, prepositions, and determiners) with words from the same class. In order to do so, we use Stanza's Universal POS tagger (Qi

et al., 2020).[2]

This process allows us to model more fine-grained syntactic categories than sampling via POS alone. For example, the substitutions for the word 'thought' in Figure 3 demonstrate how not just any verb can be substituted for any other. Instead, correct substitutions must be sensitive to subcategorization (the syntactic argument(s) required). In this case, 'thought' requires a clausal complement, which 'figured,' and 'knew' both admit. Substituting 'thought' with a verb like 'eat' would result in ungrammaticality or an altered syntactic structure.

We can denote a sentence where the word at position $j$ is replaced with word $x$ as $s\backslash(x, j)$ and the set of such sentences $S_{sub}(s, j, X)$. This is defined on the syntactic category $X$ as given in Definition 1.

$$S_{sub}(s, j, X) := \{s\backslash(x, j) | x \in X\}. \qquad (2)$$

### 3.5 Inducing trees with syntactic relations

We will now explore how to apply this newly defined set of syntactically invariant sentences to the extraction of structures from attention distributions.

Given that our relations $r \in R_{synt}$ satisfy Definition 1, if $Dep_{synt}(s, i, j) = 1$, then $\exists r \in R_{synt}$ such that it relates $w_{(i)}$ and $w_{(j)}$ and defines a syntactic category $X$ of valid substitutions;

$$if\ Dep_{synt}(s, i, j) = 1,$$
$$then\ \forall s' \in S_{sub}(s, j, X), Dep_{synt}(s', i, j) = 1.$$
$$(3)$$

Importantly, any sentence $s' \in S_{sub}(s, j, X)$ has the *same syntactic structure* as the original, $s$.

Given this basic assumption of the properties of syntactic relations in a dependency grammar, we can now propose a method of extracting syntactic structures from LLM attention distributions. Rather than applying the $MST$ algorithm on the attention distributions of a single sentence, we apply the $MST$ over an attention matrix which is derived by some algorithm, $f$ applied over the *set of attention matrices of the set of sentences created via substitution* $S_{sub}(s, i, X), \forall i \in [0, n-1]$.

$$Att_{sub}(s) = f(\{Att(s') | \forall s' \in S_{sub}(s, i, X),$$
$$i \in [0, n-1]\}) \quad (4)$$

---

[2]This is the strictest theoretical setting, however, see Experiment 3 for further discussion.

Recall that in the hypothesis represented by Equation 1, the assumption is that words in syntactic dependencies have higher attention scores than those that are not. The new hypothesis that we test in this work is that using the attention distributions of a single target sentence may reveal little about *syntactic* dependencies, which we propose must satisfy Definition 1. Instead, we use the attention patterns over a set of syntactically invariant sentences, as defined by the procedure we gave in §3.4.

Concretely, we test whether an *averaged* attention distribution over the set $S_{sub}(i, x), i \in [0, n-1]$ better reflects syntactic dependencies, i.e.

$$\forall (i, j) \in \{(a, b) | Dep_{synt}(s, a, b) = 1\},$$
$$\forall (y, z) \in \{(c, d) | Dep_{synt}(s, c, d) = 0\},$$
$$avg(a_{ij}^{s'} | \forall s' \in S_{sub}(s, i, X)) >$$
$$avg(a_{yz}^{s'} | \forall s' \in S_{sub}(s, i, X)), \quad (5)$$

and whether taking the maximum spanning tree of these averaged attention scores provides better resulting parses, $t_s = MST(Att_{sub}(s))$. Equations 1 and 5 provide a comparison between the previous work and our proposal.

Additionally, we suggest the following function $f$ for combining attention distributions between sentences: each row $i$ in the output matrix is equal to the *averaged* $i^{th}$ row of the attention distributions over the sentences which are substituted at $w_{(i)}$, i.e.

$$Att_{sub}(s)[i] =$$
$$avg(\{Att(s')[i] | \forall s' \in S_{sub}(s, i, X)\}). \quad (6)$$

We define our method, SSUD, as tree induction for a sentence $s$, which uses an attention distribution, $Att_{sub}(s)$, produced by averaging over $k$ substitutions at each position, $|S_{sub}(s, i, X)| = k$.

Our experiments in the sections below investigate whether modeling syntactic substitutability with SSUD results in the induction of better syntactic structures than using the target sentence alone. We test our hypothesis on standard datasets and long-distance subject-verb agreement constructions. SSUD is used in two different parsing setups, providing direct comparison with a previously proposed tree induction algorithm.

## 4 Datasets and Models

As in previous work (e.g. Hoover et al., 2021), we assess our method using two gold-standard En-

glish dependency parsing datasets: (1) the sentence length $\leq 10$ test split (section 23) of the Wall Street Journal portion of the Penn Treebank (Marcus et al., 1993) annotated with Stanford Dependencies (de Marneffe et al., 2006) (WSJ10; 389 sentences), and (2) the English section of the Parallel Universal Dependencies dataset annotated with Universal Dependencies (Nivre et al., 2020) (EN-PUD; 1000 sentences). Additionally, we assess our parses with Surface-Syntactic Universal Dependencies annotations (Gerdes et al., 2018; see §5.4). We use section 21 of the Penn Treebank as a validation set. We also test our method on a more difficult, long-distance subject-verb agreement dataset from Marvin and Linzen (2018) (see Experiment 2).

The language model investigated here is BERT (Devlin et al., 2019), a transformer-based language model. Specifically, we focus on the `bert-base-uncased` model, which has 12 layers with 12 self-attention heads each (110M parameters). To test generalization with respect to model size, we also use `bert-large-uncased` (336M parameters) in Experiment 1.

## 5 Does modeling syntactic substitutability increase parse accuracy?

### 5.1 Experiment 1: Setup

In **Experiment 1.1**, we induce trees over attention distributions computed by averaging all heads at a given layer. We apply our proposed method SSUD and compare the Unlabeled Undirected Attachment Score (UUAS, as in Hewitt and Manning, 2019) of trees which are induced using only the attention distributions of the target sentence, with trees resulting from applying SSUD. UUAS is calculated as the number of edges in the gold-annotated parse which are also predicted by the model, divided by the total number of edges. In this experiment, we apply SSUD with $k = 1$ (one additional sentence per word) to choose a layer. We expect SSUD to work only when syntactic information is represented.

In **Experiment 1.2**, we test the effect of SSUD by increasing the number of additional sentences used for each word, applying this on the best-performing layer from above. As syntactic substitutability is modeled using sets of sentences, the effect is expected to be greater as more appropriate substitutions are made.

In both experiments, we induce trees over the attention matrices using Prim's algorithm (Prim, 1957) which produces non-projective undirected,

unlabeled trees. This allows us to investigate the effect of modeling syntactic substitutability without making more assumptions about the directionality of the relations. Given this algorithm, the sentences which contribute the scores for all edges predicted could have been substituted at any position, including at either the head or tail of a given dependency. We make no assumptions regarding the projectivity of the resulting tree and apply this uniformly across the comparisons that we make. See Experiment 3 for SSUD with an algorithm for directed trees.

## 5.2 Experiment 1.1: Results and Discussion

For `bert-base-uncased`, the layer with the largest change in UUAS on the validation set between using the target sentence and using SSUD is Layer 10 (Appendix A). This generalizes to the results on both test sets, Table 1. With `bert-large-uncased`, we observe that Layers 17 and 18 perform the best (Table 9). As predicted, this may reflect the fact that syntactic information is more robustly represented in these layers.

For both the `base` and `large` models, our findings with regards to which layers contain retrievable syntactic information corroborate previous work. We find that Layer 10 and Layers 17 and 18 perform best for the `base` and `large` models, respectively. Previous work in constituency parsing with the same models, Kim et al. (2020), find that Layer 9 (`base`) and Layer 16 (`large`) perform best. Probing experiments have also previously shown that constituency information occurs before dependency information in BERT's representations (Tenney et al., 2019).

In Experiment 1.2, we further investigate the effect of SSUD by providing more substitutions at each word (increasing $k$). For the following experiments, we use Layer 10 of the `base-uncased` model and Layer 17 of `bert-large-uncased`.

## 5.3 Experiment 1.2: Results and Discussion

Table 2 provides the results as the number of substitutions is increased. Improvements are seen for both models on WSJ-10 and EN-PUD. There is a marginally larger increase for the EN-PUD dataset which has sentences of average length 17, compared to sentences of $\leq 10$ for WSJ-10. The monotonic increase in UUAS as more sentences are added suggests that our method of modeling substitutability using sentence substitutions is an effective constraint for distilling syntactic information from models. It also suggests that the syntactic

| | UUAS | | | | | |
|---|---|---|---|---|---|---|
| | WSJ10 | | | EN-PUD | | |
| Layer | T. | $k=1$ | $\Delta$ | T. | $k=1$ | $\Delta$ |
| 6 | 57.3 | 57.3 | 0.0 | 44.8 | 44.8 | 0.0 |
| 7 | 56.3 | 56.4 | 0.1 | 44.2 | 44.1 | -0.1 |
| 8 | 56.0 | 56.1 | 0.1 | 43.2 | 43.2 | 0.0 |
| 9 | 55.9 | 55.8 | -0.1 | 43.9 | 44.0 | 0.1 |
| 10 | 55.7 | 56.8 | **1.1** | 44.3 | 44.7 | **0.4** |

Table 1: UUAS scores on WSJ10 and EN-PUD (`bert-base-uncased`). SSUD $k=1$ compared with only using target sentence (*T.*).

| | `bert-base-uncased` (UUAS) | | | | |
|---|---|---|---|---|---|
| | T. | $k=1$ | $k=3$ | $k=5$ | $k=10$ |
| WSJ10 | 55.7 | 56.8 | 57.0 | 57.3 | **57.6** |
| EN-PUD | 44.3 | 44.7 | 45.6 | 46.2 | **46.4** |
| | `bert-large-uncased` (UUAS) | | | | |
| WSJ10 | 56.1 | 56.5 | 56.7 | 56.7 | **57.2** |
| EN-PUD | 45.5 | 45.8 | 46.2 | 46.6 | **47.0** |

Table 2: Results on WSJ-10 and EN-PUD for `bert-base-uncased` (Layer 10) and `bert-large-uncased` (Layer 17). Comparison between using the target sentence alone (*T.*), and SSUD, with an increasing number of substitutions, $k = 1, 3, 5, 10$.

| Method | WSJ10 UUAS | EN-PUD UUAS | UAS |
|---|---|---|---|
| Ours – Experiment 1 | 57.6 | 46.4 | – |
| Zhang and Hashimoto (2021) | 58.74 | –[§] | – |
| Hoover et al. (2021)* | 53.- | 43.-[¶] | – |
| Klein and Manning (2004)[†] | 55.91 | – | – |
| Limisiewicz et al. (2020)[‡] | – | 59.9 | 52.8 |
| Ours – Experiment 3 | – | 62.0 | 54.5 |

Table 3: Results from Experiments 1 and 3 are reported with comparisons to previously proposed methods (best non-projective scores from `bert-base-uncased`). [§]not included due to computation time; [†]reported in Zhang and Hashimoto (2021); [¶]results are from multilingual BERT. *'abs CPMI' trees experiment in the original paper; [‡] '1000 selection sentences' experiment in the original paper.

representations intrinsic to the model are robust to substitution, allowing SSUD to disentangle syntactic information better than using the target sentence alone. We provide further analysis of the results from the `bert-base-uncased` model.

In Table 3, we provide comparisons to other previously proposed methods. We see that SSUD is competitive with other reported UUAS scores. We suggest that even though our method does not achieve state-of-the-art scores, they are comparable and the performance increases are reliable. In the following experiment, we provide more fine-

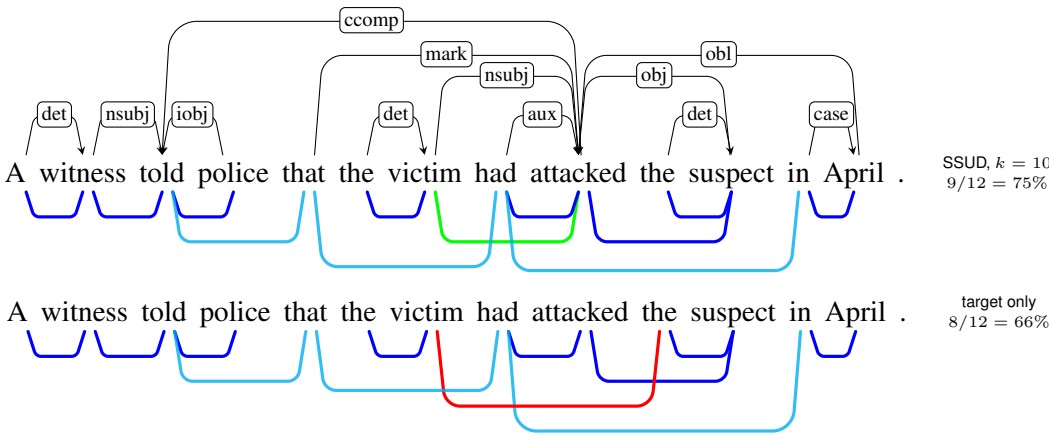

Figure 4: Dependency parse which compares the result of SSUD and using the target sentence alone. In the SSUD parse, the noun, 'victim' is accurately attached to its verb 'attacked' (in green). Without SSUD, 'victim' is attached to the determiner of 'suspect,' perhaps due to lexical similarity (in red). Dark blue edges match the gold-standard parse. Light blue edges demonstrate how induced edges can differ, but still be syntactically informative.

|  | EN-PUD (UUAS) | | |
|---|---|---|---|
|  | T. | $k = 10$ | $\Delta$ |
| UD (Nivre et al., 2020) | 44.3 | 46.4 | 2.1 |
| SUD (Gerdes et al., 2018) | 56.0 | 59.0 | **3.0** |

Table 4: UUAS scores for Universal Dependencies annotations (UD) and Surface-Syntactic Universal Dependencies annotations (SUD) on sentences from EN-PUD (`bert-base-uncased`). Comparison between using the target sentence alone (T.) and SSUD, $k = 10$.

grained comparisons to previous methods by looking at more challenging syntactic constructions.

In Figure 4, we provide an example of a parse tree induced via SSUD and one which only uses the target sentence. We have labeled some edges which differ from the UD annotation, but which are still syntactically informative, e.g. predicting an edge from the matrix verb 'told' to the complementizer 'that,' rather than to the main verb of the embedded clause (see Appendix C for more example parses). Cases such as these demonstrate that specific choices from annotation schemata can artificially lower the resulting UUAS scores. We now test this observation quantitatively.

### 5.4 Results and Discussion: Which syntactic formalism do SSUD parses align with?

In order to compare differences resulting from annotation choices, we rescore the EN-PUD trees induced via SSUD on a different syntactic formalism. Specifically, we choose the Surface-Syntactic UD formalism (SUD) (Gerdes et al., 2018), which differs from UD mainly in one respect: function

words are treated as heads of relations in SUD. For example, in SUD, a verb with a clausal complement would be attached to a complementizer as we noted in our qualitative analysis above.

Table 4 shows that SSUD parses receive higher scores on SUD (59.0 vs. 46.4 UUAS) and that using our method on SUD provides a larger improvement (+3.0pts vs. +2.1pts). We also find differences when looking at recall scores for specific relations which differ between the formalisms (see Appendix B for full relation-wise results). For example, two relations which are annotated with content words as heads in UD, obl and ccomp, both receive low recall: 2.3% and 11.1%, respectively. In contrast, the two SUD relations which subsume these two relations, comp:obj and comp:obl, achieve much higher recall: 57.56% and 79.3%.

This result supports our qualitative analysis in the previous section, however, Kulmizev et al. (2020) come to the opposite conclusion when assessing the preferences of BERT for the same two annotation schemata. Since they use a trained probe to induce parses, perhaps different kinds of linguistic information are being recovered via our two distinct methods. We leave further analysis of this to future work.

## 6 Does SSUD help with harder syntactic constructions?

### 6.1 Experiment 2: Dataset and Setup

In the previous experiments, we provided results for our method applied to standard parsing datasets. In this experiment, we use data from Marvin and

Linzen (2018) to control for the syntactic structures being evaluated. Specifically, we look at more challenging long-distance subject-verb agreement constructions which have been used to investigate hierarchically-dependent behaviour in language models. The reasoning here is that models using linear heuristics such as the distance between a noun and a verb would mistakenly assign closer nouns as nominal subjects. We reframe this task and investigate whether the tree-induction methods are able to accurately predict an edge between the subject and verb. We report a correctly predicted edge as either between the subject's determiner or head noun and the verb.

We sample 1000 sentences from 2 templates used in Marvin and Linzen (2018): agreement across an object relative clause (e.g. 'The pilot [that the minister likes] cooks.') and agreement across a subject relative clause (e.g. 'The customer [that hates the skater] swims.'). We include only non-copular verbs to control for any possible differences in syntactic representation.

We evaluate SSUD on this task and provide results from applying Zhang and Hashimoto (2021)'s conditional MI method, which performs better than ours in the previous task, for comparison.

## 6.2 Experiment 2: Results and Discussion

The results are shown in Table 5, with an improvement in edge recall for SSUD as $k$ is increased. This further corroborates our observations from the previous experiments on standard datasets, and the increase of 8.4 points for the object relative clauses, and 8.3 points for subject relative clauses are promising. Comparing our results to those from applying Zhang and Hashimoto (2021)'s method are promising as well. SSUD outperforms theirs on both object (+70.6pts) and subject relative clauses (+61.1pts). A future extension to this experiment which could improve the interpretability of model mechanisms is to compare the results of syntactic structure induction with an evaluation of model *behaviour* (i.e. does a correctly predicted edge lead to correct agreement).

## 7 Does SSUD generalize?

### 7.1 Experiment 3: Setup

In this experiment, we test whether SSUD robustly improves syntactic dependency parsing by applying it to a different parsing algorithm proposed by Limisiewicz et al. (2020) for extracting directed

|  | | Object Relative Clause (recall) | | | |
| Method | *T.* | $k=1$ | $k=3$ | $k=5$ | $k=10$ |
| --- | --- | --- | --- | --- | --- |
| Ours | 71.1 | 71.2 | 72.4 | 75.3 | **79.5** |
| Z + H | 8.9 | – | – | – | – |
|  | | Subject Relative Clause (recall) | | | |
| Ours | 54.7 | 57.9 | 60.1 | 61.2 | **63.0** |
| Z + H | 1.9 | – | – | – | – |

Table 5: Results on subject-verb edge prediction. We compare using the target sentence alone (*T.*) with using SSUD $k = 1, 3, 5, 10$. For comparison, scores for conditional MI trees averaged over 3 seeds using only the target sentence are reported (Z+H), as proposed in Zhang and Hashimoto (2021).

dependency trees from attention distributions. We can directly test the effect of SSUD simply by using SSUD-processed attention matrices whenever attention matrices are used in the original algorithm.

This method involves a key additional step of selecting syntactically informative attention heads based on UD relations before inducing syntactic trees. This process requires supervision from gold-standard parses but, as such, provides an 'upper bound' of how much UD-like syntactic structure can be retrieved from BERT. Heads are selected for both the dependent-to-parent and parent-to-dependent directions for each relation. As with previous experiments, we compare SSUD to using the target sentence only, and evaluate both steps of the algorithm: (i) are the chosen heads more accurate for the UD relations considered, and (ii) does SSUD improve the induced syntactic trees? We constrain our method and use the same resources and models as the original algorithm[3] and do not use POS information. We test the best-performing method in their paper which uses 1000 selection sentences. Following the original paper, directed labeled and unlabeled trees are induced and unlabeled attachment scores and labeled attachment scores on the EN-PUD dataset are used for evaluation.

### 7.2 Experiment 3: Results and Discussion

The results of the experiment are summarized in Table 6. For **head selection**, SSUD outperforms using the target sentence alone in all but 3 relations: aux (-0.2pts) and amod (-0.1pts) in the dependent-to-parent direction, and nummod (-0.8pts) in the parent-to-dependent direction. There is no relation that using SSUD does not improve for at least one

---

[3]https://github.com/tomlimi/BERTHeadEnsembles.

| Dependent-to-Parent Head Selection Accuracy | | | | | |
|---|---|---|---|---|---|
| Label | $T.$ | $k=1$ | $k=3$ | $k=5$ | $\Delta$(SSUD, $T.$) |
| nsubj | 63.8 | 65.8 | 67.0 | **68.2** | 4.4 |
| obj | 91.1 | 92.7 | **93.9** | **93.9** | 2.8 |
| det | 97.3 | **97.4** | 96.8 | 95.7 | 0.1 |
| case | 88.0 | 88.0 | **88.2** | 87.9 | 0.2 |
| Tree Induction Scores | | | | | |
| Metric | $T.$ | $k=1$ | $k=3$ | $k=5$ | $\Delta$(SSUD, $T.$) |
| UAS | 52.8 | 53.7 | **54.5** | 54 | 1.7 |
| LAS | 22.5 | 25.6 | **26.3** | 22 | 3.8 |

Table 6: Results on selected non-clausal relations used for head selection in the dependent to parent direction, full results in Appendix D. Unlabeled (UAS) and labeled (LAS) scores are reported as well. Using only the target sentence ($T.$) is equivalent to the original algorithm; results for SSUD ($k = 1, 3, 5$) are provided.

of the directions. For **tree induction**, we see that SSUD $k = 3$ provides the highest improvements with increases in both the unlabeled (+1.7pts) and labeled (+3.8pts) attachment scores. Unlike in the previous experiments, increasing the number of substitutions does not monotonically increase parsing accuracy. We analyze this effect below.

As stated in the setup, we wished to maintain the resources and models used in the original paper and as such do not use POS information here. This in turn leads to a split between lexical and functional categories. Increasing the number of substitutions for closed categories like determiners can lead to a decrease in performance if the number of possible substitutions is exceeded. Quantitative results reveal this is the case: for example, as more substitutions are used, correctly labeling the det relation falls from 38.6 ($k = 3$) to 7.9 ($k = 5$). The head selection accuracy patterns in Table 6 reflect this as well. Interestingly, at $k = 5$ the model incorrectly labels det as case 48.2% of the time. However, when considering a relation with *open* class words like obj, SSUD $k = 5$ labels obj correctly with 36.6 recall, outperforming $T.$ by 11.6pts. We refer the reader to Appendix D for full results. While we only explore a static number of substitutions here, future work may find that a dynamic number of substitutions leads to further improvements.

Overall, the results for Experiment 3 show that SSUD leads to gains over the original algorithm and effectively distills more syntactic information even when used in a different setup.

## 8 Conclusion

The results across the three experiments show that there is merit to modeling the property of syntactic substitutability when inducing syntactic depen-

dency structures. Indeed attention distributions do capture a surprising amount of syntactic information, despite never being trained to do so. With substitutability as a constraint, we can better make use of this information for unsupervised parsing and better understand the extent to which this property is learned by attention mechanisms. Our results also suggest that previous probing results on attention mechanisms using single sentences may underestimate the amount of syntactic information present.

We show that SSUD effectively transfers to new parsing setups and to different datasets. A potential next step is to use this method cross-linguistically in order to better understand the representation of different languages within multilingual models. In addition to uses for NLP and computational linguistics, these interpretable, model-intrinsic structures may provide a new source of data for work in theoretical linguistics and syntactic annotation, and as further confirmation of the emergence of syntax-like structures in neural network language models.

## Ethics Statement

While there are few ethical problems linked to the methodology explored in this paper, there remain those more generally associated with large language models, including concerns of privacy and learned biases. A better understanding of linguistic mechanisms in models may lead to improvements in this domain as well.

## Limitations

In this paper, we focus only on English data. This is a limitation of the work because English is a language with relatively strict word order and does not morphologically mark most grammatical relations. This may present a challenge when this method is used in other languages as substituting a given word in a sentence may affect morphological marking on other words in the sentence, but we hope large-scale training of BERT-like models may circumvent some of these problems. Theoretically, we can capture this by assuming more fine-grained categories which do not differ in this morphology (see Dehouck and Gómez-Rodríguez (2020) for a discussion).

Another limitation of this study is that we only study BERT models trained on English. This has a twofold effect: (1) there is nothing stopping the attention mechanisms of a different model from

storing syntactic information differently, and (2) as previously mentioned, English has a relatively rigid word order which may already carry much information about the syntactic relations between words. Compared to a language with more word orders like Japanese or German, it is not unimaginable that attention mechanisms learn to track syntactic relations differently. Addressing the small number of models studied here, we suggest that the main contribution of this method, that syntactic substitutability can help extract syntax from models, is one which is defined agnostically of the specific model. As such, the results of applying this method to different architectures may in fact be informative for the interpretation of their internal mechanisms as well.

The domains explored in this paper are limited: the WSJ10 dataset features sentences from news articles in the Wall Street Journal (Marcus et al., 1993), and the EN-PUD dataset are sentences from news and Wikipedia articles (Nivre et al., 2020).

## Acknowledgements

We thank members of SR's research group and the Montréal Computational & Quantitative Linguistics Lab for their helpful comments. JJ acknowledges the support of a Natural Sciences and Engineering Research Council of Canada (NSERC) USRA [funding reference number 539633]. SR acknowledges the support of the NSERC Discovery Grant program.

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

## A Experiment 1.1: Validation set results

We report Experiment 1.1 results for the validation set. Layer 10, which shows the largest increase, was chosen for Experiments 2 and 3.

| | UUAS | | |
|---|---|---|---|
| | WSJ Section 21 (validation) | | |
| Layer | $T.$ | $k=1$ | $\Delta$ |
| 6 | 52.5 | 52.7 | 0.2 |
| 7 | 51.4 | 51.7 | 0.3 |
| 8 | 50.7 | 50.8 | 0.1 |
| 9 | 52.1 | 52.0 | -0.1 |
| 10 | 49.3 | 49.9 | 0.6 |

Table 7: UUAS scores on validation set. SSUD $k=1$ compared with only using target sentence (*T.*).

## B Experiment 1.2: What does SSUD improve?

In this section, we provide additional results from Experiment 1.2.

One noted qualitative improvement due to SSUD is a decrease in non-syntactic edges being predicted between words which are semantically similar, but not syntactically dependent. This provides a potential explanation for the quantitative results in Table 8 which shows increases in the *recall* of adjacent edges and *precision* of non-adjacent edges. This suggests that fewer incorrect non-adjacent edges are being predicted (i.e. ones predicted due to lexical similarity) and more correct adjacent edges (i.e. closer dependencies that are not necessarily semantically dependent or similar). See Figure 4 for an example and Appendix C for further discussion of this and more examples.

| | | WSJ10 | | | | |
|---|---|---|---|---|---|---|
| | | $T.$ | $k=1$ | $k=3$ | $k=5$ | $k=10$ | $\Delta_{10-T.}$ |
| Adjacent | Rec. | 73.8 | 76.3 | 76.8 | 77.7 | 77.4 | **3.6** |
| | Prec. | 67.7 | 68.3 | 68 | 68.3 | 67.8 | 0.1 |
| Non-Adj. | Rec. | 34.5 | 34 | 33.7 | 33.3 | 34.2 | -0.3 |
| | Prec. | 38.5 | 39.3 | 39.7 | 39.7 | 41.1 | **2.6** |
| | | EN-PUD | | | | |
| | | $T.$ | $k=1$ | $k=3$ | $k=5$ | $k=10$ | $\Delta_{10-T.}$ |
| Adjacent | Rec. | 71.7 | 73 | 74.7 | 75.7 | 76.1 | **4.4** |
| | Prec. | 55.5 | 55.6 | 55.9 | 56 | 56 | 0.5 |
| Non-Adj. | Rec. | 24.5 | 24.2 | 24.6 | 24.8 | 24.9 | 0.4 |
| | Prec. | 31.1 | 31.3 | 32.4 | 33.3 | 33.7 | **2.6** |

Table 8: Adjacent and non-adjacent edge recall and precision results on WSJ10 and EN-PUD. Results are reported across an increasing number of substitutions, $k = 1, 3, 5, 10$ for Layer 10.

In Tables 10 and 11, we report the relation-wise results for both datasets. We see similar improvements in both the WSJ-10 dataset and the EN-PUD

dataset with SSUD: +2.0/+1.7 for nsubj, +4.0/+1.5 for dobj/obj, among other parallels. Again, note that annotation schemata may be defined differently from the edges induced, even if they are syntactically informative. For example, the ccomp (clausal complement) relation, which links a verb or adjective with a dependent clause, has a relatively low 11.1% recall in the EN-PUD dataset. Looking at an example like Figure 5 in Appendix Section C begins to show why this may be: the verb 'told' is linked with the complementizer 'that,' rather than the main verb of the embedded clause 'attacked,' as is done in the UD schema.

### B.1 Surface-Syntactic UD

In Table 12, we provide full results on the English Parallel Surface-Syntactic UD annotated dataset. As discussed, this formalism treats functional words like complementizers and prepositions as the heads of dependencies. This is the pattern that we have qualitatively noted in SSUD derived parses. The results show that SSUD favours SUD, achieving a 57.6% recall on the SUD comp:obj and 79.3% on the comp:obl relation which encompass the UD ccomp relation mentioned above. The comparison is not robust as those two relations also include other UD complement relations, though the SUD scores are reliably higher.

## C Experiment 1.2: Example parses

In Figures 5, 6, and 7, we provide some example parses comparing our method, and those induced using conditional MI with Zhang and Hashimoto (2021)'s method, seed = 1. The errors and improvements we note relate to the results discussed in Appendix B, including a decrease of non-syntactic edges predicted due to semantic similarity, and specific differences between induced trees and annotation-dependent choices.

## D Experiment 3: Additional results

We provide full per relation results for Experiment 3 in this section. Results for head selection accuracy for both the dependent-to-parent and parent-to-dependent direction are provided in Tables 13 and 14, respectively. As reviewed in §7.2, SSUD outperforms the original algorithm with just the target sentence on all relations except on aux (-0.2pts) and amod (-0.1pts) in the dependent-to-parent direction, and nummod (-0.8pts) in the parent-to-dependent direction.

Per relation results on unlabeled and labeled trees are also provided in Tables 15 and 16. The recall scores for UAS are calculated as the number of edges predicted correctly for each relation, while for LAS both the edge and correct label must be predicted. As reviewed in §7.2 and Table 6, SSUD $k = 3$ provides the best results, and much of the decrease between $k = 3$ and $k = 5$ can be attributed to relations with closed class lexical items like det, while open class relations like obj and subj remain relatively stable or show improvements as substitutions are increased.

## E   Experiment 1: `bert-large-uncased`

| | UUAS | | | | | |
| | WSJ10 | | | EN-PUD | | |
| Layer | T. | $k = 1$ | $\Delta$ | T. | $k = 1$ | $\Delta$ |
|---|---|---|---|---|---|---|
| 16 | 54.6 | 54.8 | 0.2 | 43.6 | 43.8 | 0.2 |
| 17 | 56.1 | 56.5 | 0.4 | 45.5 | 45.8 | 0.3 |
| 18 | 53.5 | 54.3 | 0.8 | 41.5 | 41.9 | 0.4 |

Table 9:  UUAS scores on WSJ10 and EN-PUD (`bert-large-uncased`). SSUD $k = 1$ compared with only using target sentence (*T.*).

In Table 9, we provide full results for Experiment 1 for the `bert-large-uncased` model. The best-performing layer corresponds with previous results from the literature about the locus of retrievable syntactic information in pretrained BERT models (e.g. Kim et al., 2020).

## F   Compute and package versions

The SSUD experiments can be reproduced with a GPU with 2GB of memory, and a CPU with 24GB of memory. Experiments each run in 7hrs, on an RTX8000.  Experiments comparing our method with Zhang and Hashimoto (2021) used a GPU with 24GB of memory, and CPU with 100GB of memory. These experiments ran in 10hrs on an RTX8000 GPU. Packages: Stanza (1.4.0); networkx (1.22.4); numpy (1.22.4); transformers (4.19.2); torch (1.11.0).

Experiments involving the algorithm proposed in Limisiewicz et al. (2020) used a GPU with 24GB of memory, and a CPU with 128GB. These experiments ran in 10hrs, on an RTX8000. We direct the reader to the original repository for packages used therein.

## G   Datasets and licenses

The Stanza (Qi et al., 2020) package was used as intended, under the Apache License, Version 2.0.

The datasets were used as intended, as established by previous work such as Klein and Manning (2004) and Hoover et al. (2021). EN-PUD is part of the publicly available Parallel Universal Dependencies treebanks. Surface-Syntactic UD treebanks are also available publicly. The WSJ datasets were acquired through a Not-For-Profit, Standard Linguistic Data Consortium licence. The data are from published news, and Wikipedia sources.

| | Recall (WSJ10) | |
|---|---|---|
| SD Relation | T. | k = 10 |
| acomp | 64.3 | 78.6 |
| advcl | 0.0 | 0.0 |
| advmod | 58.2 | 60.3 |
| amod | 72.7 | 74.8 |
| appos | 65.0 | 65.0 |
| aux | 45.0 | 43.3 |
| auxpass | 70.8 | 79.2 |
| cc | 37.0 | 38.9 |
| ccomp | 5.9 | 5.9 |
| conj | 51.5 | 51.5 |
| cop | 34.8 | 44.9 |
| csubj | 50.0 | 50.0 |
| dep | 45.5 | 48.5 |
| det | 76.7 | 75.7 |
| discourse | 60.0 | 40.0 |
| dobj | 58.2 | 62.2 |
| expl | 100.0 | 85.7 |
| iobj | 50.0 | 50.0 |
| mark | 25.0 | 25.0 |
| neg | 23.4 | 27.7 |
| nn | 65.4 | 67.5 |
| npadvmod | 25.0 | 50.0 |
| nsubj | 45.2 | 47.2 |
| nsubjpass | 25.0 | 20.8 |
| num | 67.3 | 67.3 |
| number | 85.2 | 85.2 |
| parataxis | 0.0 | 0.0 |
| pcomp | 100.0 | 100.0 |
| pobj | 63.7 | 63.7 |
| poss | 45.7 | 48.6 |
| possessive | 68.8 | 87.5 |
| preconj | 0.0 | 0.0 |
| predet | 100.0 | 0.0 |
| prep | 58.2 | 60.5 |
| prt | 100.0 | 100.0 |
| quantmod | 57.1 | 57.1 |
| rcmod | 0.0 | 0.0 |
| tmod | 33.3 | 55.6 |
| vmod | 70.0 | 60.0 |
| xcomp | 14.7 | 23.5 |

Table 10: Experiment 1.2: Per relation results on WSJ10, annotated with Stanford Dependencies, comparing target only (T.), and SSUD, k = 10. Note: recall of a relation may be lower if the induced trees differ from annotation schemata, despite syntactic relevance.

| | Recall (EN-PUD) | |
|---|---|---|
| UD Relation | T. | k = 10 |
| acl | 35.2 | 34.7 |
| acl:relcl | 21.3 | 24.2 |
| advcl | 6.5 | 5.5 |
| advmod | 49.6 | 53.4 |
| amod | 68.7 | 72.9 |
| appos | 27.3 | 23.1 |
| aux | 32.9 | 33.4 |
| aux:pass | 71.2 | 74.8 |
| case | 55.1 | 59.9 |
| cc | 32.6 | 33.8 |
| cc:preconj | 0.0 | 9.1 |
| ccomp | 8.1 | 11.1 |
| compound | 79.1 | 84.6 |
| compound:prt | 95.7 | 98.6 |
| conj | 31.1 | 28.2 |
| cop | 30.4 | 33.2 |
| csubj | 11.1 | 11.1 |
| csubj:pass | 0.0 | 0.0 |
| dep | 0.0 | 0.0 |
| det | 68.5 | 72.9 |
| det:predet | 33.3 | 33.3 |
| discourse | 0.0 | 0.0 |
| dislocated | 0.0 | 0.0 |
| expl | 69.4 | 66.1 |
| fixed | 85.4 | 85.4 |
| flat | 78.3 | 79.1 |
| goeswith | 100.0 | 100.0 |
| iobj | 40.0 | 40.0 |
| mark | 52.1 | 51.7 |
| nmod | 12.3 | 10.2 |
| nmod:npmod | 63.2 | 68.4 |
| nmod:poss | 40.0 | 41.4 |
| nmod:tmod | 38.5 | 43.6 |
| nsubj | 28.3 | 30.0 |
| nsubj:pass | 26.4 | 25.9 |
| nummod | 73.2 | 74.0 |
| obj | 41.8 | 43.3 |
| obl | 3.1 | 2.3 |
| obl:npmod | 50.0 | 45.0 |
| obl:tmod | 22.2 | 11.1 |
| orphan | 14.3 | 14.3 |
| parataxis | 6.2 | 7.2 |
| reparandum | 0.0 | 0.0 |
| vocative | 0.0 | 100.0 |
| xcomp | 19.9 | 21.4 |

Table 11: Experiment 1.2: Per relation results on EN-PUD, annotated with Universal Dependencies, comparing target only (T.), and SSUD, k = 10. Note: recall of a relation may be lower if the induced trees differ from annotation schemata, despite syntactic relevance.

|                | Recall (EN-PUD) | |
| --- | --- | --- |
| SUD Relation | *T.* | k=10 |
| appos | 28.7 | 24.5 |
| cc | 39.4 | 40.6 |
| cc@preconj | 0.0 | 9.1 |
| comp:aux | 50.5 | 51.0 |
| comp:aux@pass | 71.5 | 75.2 |
| comp:obj | 54.0 | 57.6 |
| comp:obl | 76.5 | 79.3 |
| comp:pred | 38.7 | 42.2 |
| comp@expl | 66.7 | 61.5 |
| compound | 79.1 | 84.6 |
| compound@prt | 95.7 | 98.6 |
| conj | 36.2 | 35.3 |
| conj@emb | 50.0 | 60.0 |
| det | 68.5 | 72.9 |
| det@predet | 33.3 | 33.3 |
| discourse | 100.0 | 100.0 |
| dislocated | 0.0 | 0.0 |
| flat | 86.5 | 87.4 |
| goeswith | 100.0 | 100.0 |
| mod | 61.1 | 64.4 |
| mod@npmod | 61.1 | 66.7 |
| mod@poss | 49.1 | 52.1 |
| mod@relcl | 13.7 | 15.2 |
| mod@tmod | 43.6 | 46.2 |
| orphan | 0.0 | 0.0 |
| parataxis | 12.4 | 12.4 |
| reparandum | 0.0 | 0.0 |
| subj | 34.1 | 36.1 |
| subj@pass | 23.6 | 26.0 |
| udep | 64.6 | 67.9 |
| udep@npmod | 52.4 | 47.6 |
| udep@poss | 45.1 | 43.1 |
| udep@tmod | 33.3 | 22.2 |
| unk | 85.7 | 86.7 |
| unk@expl | 73.9 | 73.9 |
| vocative | 0.0 | 100.0 |

Table 12: Experiment 1.2: Per relation results on EN-PUD, annotated with Surface-Syntactic Universal Dependencies, comparing target only (*T.*), and SSUD, $k = 10$. Note: recall of a relation may be lower if the induced trees differ from annotation schemata, despite syntactic relevance.

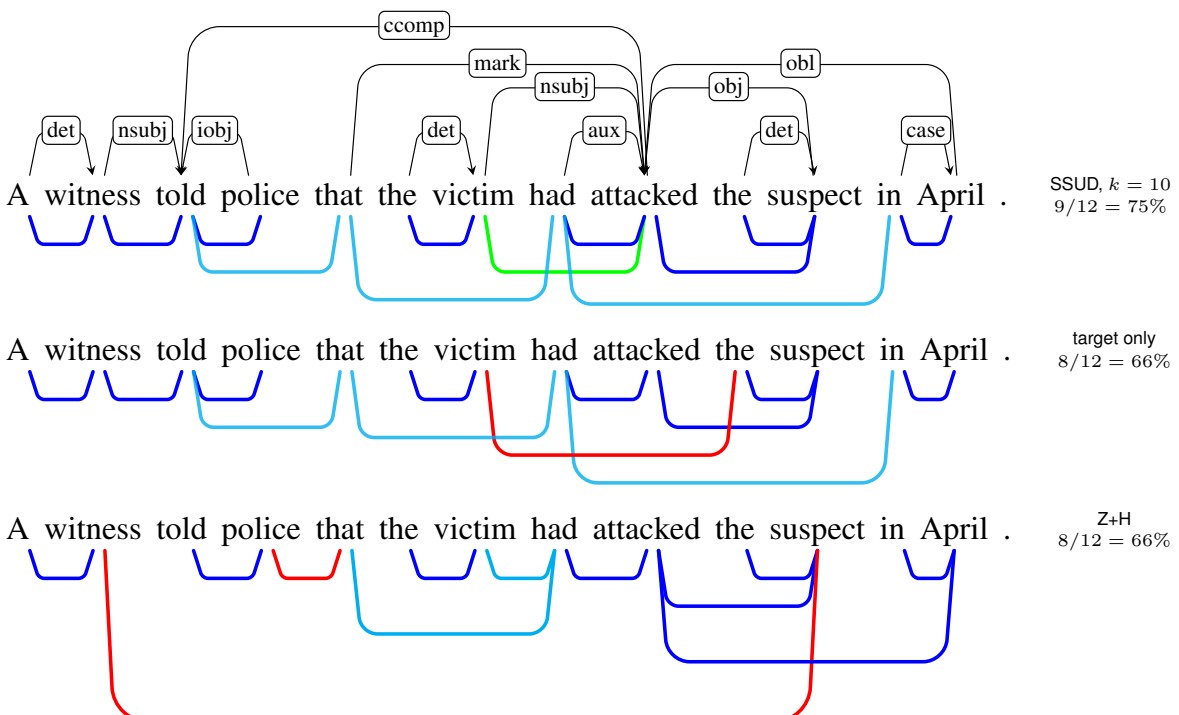

Figure 5: This example was presented in Figure 4 above. The conditional MI parse predicts an edge between 'suspect' and 'witness,' perhaps for their salience to the domain of the sentence – 'witness' is no longer attached to its verb 'told.' This is perhaps similar to the edge between 'victim' and 'the [suspect]' in the target-only parse, where 'victim' is no longer attached to the verb 'attacked'.

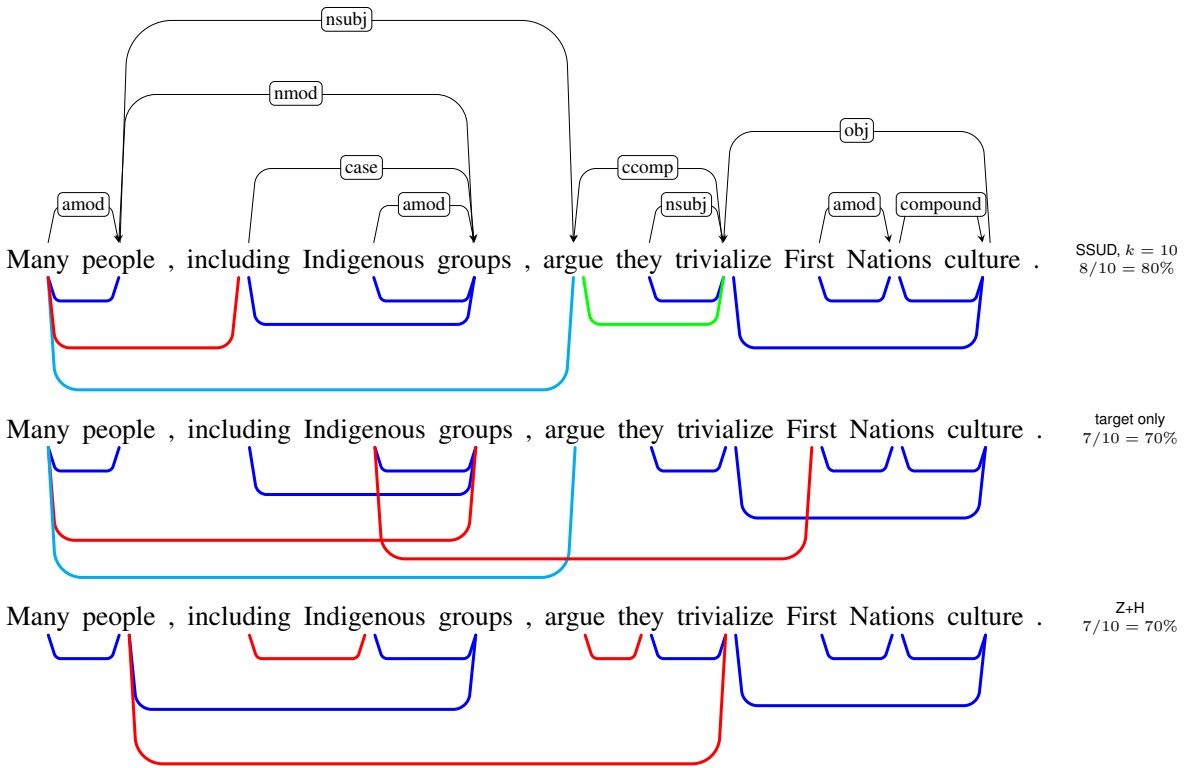

Figure 6: The edge between 'Indigenous' and 'First' in the target-only parse shows edge-prediction errors where words are linked with other semantically similar words, rather than syntactically dependent ones. An edge is also predicted between 'people' and 'trivialize' in the conditional MI parse represents an incorrect argument structure, which should be more like the SSUD parse where the noun phrase 'many people' is connected to 'argue'. We see in the SSUD parse an example where the choice of headedness in the noun phrase artificially lowers the UUAS score. Qualitatively, the nominal subject of the verb 'argue' is correctly attached. The clausal argument structure is also improved: 'argue' in the SSUD parse correctly attaches to its clausal complement at the verb 'trivialize.'

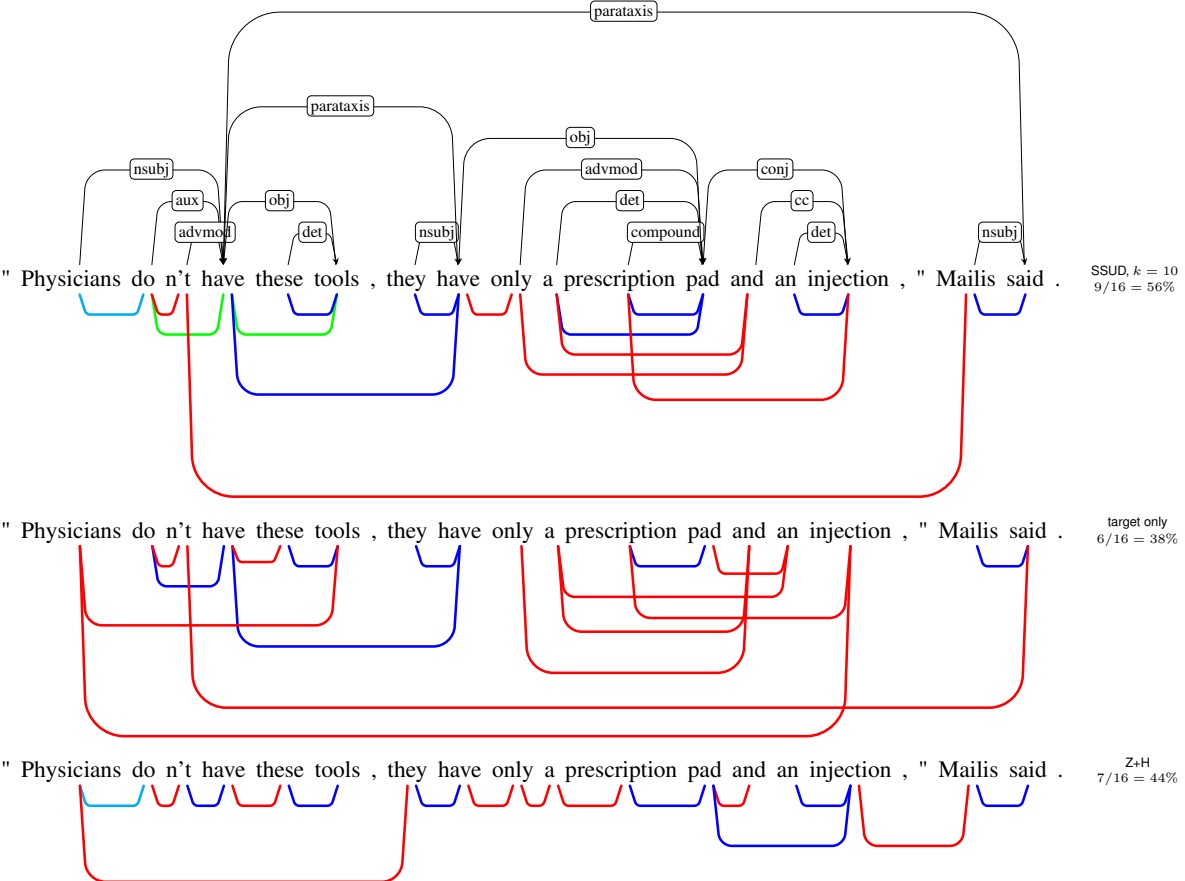

Figure 7: In the target-only parse, we again see the effect of semantic similarity on the induced parse. 'Physicians' is connected to both 'tools' and 'injections', rather than either the auxiliary verb 'do' or the main verb 'have.' This is resolved in both the SSUD and conditional MI parses. A weakness of the SSUD parse here is the lack of clarity in conjunctions, where the induced structure of 'a prescription pad and an injection' does not appear to be adequately clarified. The subtree containing this in the SSUD parse, rooted by the quantifier 'only,' is however correctly attached to the verb 'have,' unlike the target-only parse.

| d2p Head Selection | | | |
| --- | --- | --- | --- |
| UD Relation | $T.$ | $k=1$ | $k=3$ | $k=5$ |
| acl | 53.2 | 55.7 | 55.4 | 55.4 |
| advcl | 40.6 | 41.6 | 44.7 | 43.7 |
| cc | 67.0 | 68.0 | 67.5 | 68.2 |
| csubj | 56.7 | 53.3 | 56.7 | 56.7 |
| parataxis | 28.9 | 29.9 | 26.8 | 29.9 |
| amod | 94.3 | 94.1 | 94.2 | 94.1 |
| advmod | 65.5 | 66.0 | 66.7 | 67.5 |
| aux | 93.6 | 93.4 | 92.5 | 92.5 |
| compound | 86.9 | 87.4 | 88.8 | 87.6 |
| conj | 62.3 | 64.7 | 66.2 | 67.2 |
| det | 97.3 | 97.4 | 96.8 | 95.7 |
| nmod | 49.2 | 48.2 | 49.3 | 49.5 |
| nummod | 85.8 | 86.2 | 87.4 | 86.6 |
| obj | 91.1 | 92.7 | 93.9 | 93.9 |
| subj | 63.8 | 65.8 | 67.0 | 68.2 |
| case | 88.0 | 88.0 | 88.1 | 87.9 |
| mark | 75.1 | 75.3 | 75.5 | 75.7 |

Table 13: Experiment 3: Dependent-to-Parent head selection accuracy results

| p2d Head Selection | | | |
| --- | --- | --- | --- |
| UD Relation | $T.$ | k=1 | k=3 | k=5 |
| acl | 52.5 | 54.0 | 54.7 | 53.5 |
| advcl | 26.6 | 33.1 | 38.9 | 39.9 |
| cc | 58.4 | 59.9 | 59.6 | 58.1 |
| csubj | 46.7 | 50.0 | 40.0 | 36.7 |
| parataxis | 28.9 | 26.8 | 32.0 | 28.9 |
| amod | 79.7 | 80.6 | 80.7 | 80.9 |
| advmod | 64.9 | 66.0 | 64.7 | 65.1 |
| aux | 82.5 | 82.6 | 82.7 | 82.5 |
| compound | 82.4 | 83.0 | 85.3 | 84.7 |
| conj | 47.6 | 48.4 | 50.9 | 51.4 |
| det | 70.7 | 72.4 | 72.7 | 72.5 |
| nmod | 55.7 | 56.1 | 56.5 | 56.7 |
| nummod | 74.0 | 66.1 | 72.4 | 73.2 |
| obj | 81.0 | 82.7 | 84.4 | 84.8 |
| subj | 73.7 | 74.4 | 75.9 | 75.9 |
| case | 68.8 | 69.7 | 71.0 | 71.3 |
| mark | 66.3 | 67.0 | 60.9 | 62.2 |

Table 14: Experiment 3: Parent-to-Dependent head selection accuracy results

| Recall (EN-PUD) | | | |
| --- | --- | --- | --- |
| UD Relation | $T.$ | 1 | 3 | 5 |
| acl | 2.1 | 1.0 | 1.5 | 3.6 |
| acl:relcl | 1.0 | 1.0 | 1.4 | 2.4 |
| advcl | 1.0 | 0.7 | 1.4 | 2.4 |
| advmod | 57.9 | 59.6 | 59.0 | 59.6 |
| amod | 91.1 | 92.2 | 92.6 | 91.9 |
| appos | 6.3 | 4.2 | 6.3 | 5.6 |
| aux | 82.7 | 83.9 | 84.2 | 82.2 |
| aux:pass | 100.0 | 99.6 | 99.3 | 99.6 |
| case | 81.0 | 82.4 | 83.4 | 81.8 |
| cc | 60.5 | 61.7 | 64.8 | 62.4 |
| cc:preconj | 27.3 | 27.3 | 54.5 | 45.5 |
| ccomp | 0.0 | 0.0 | 0.0 | 0.0 |
| compound | 89.0 | 89.8 | 90.0 | 89.6 |
| compound:prt | 7.1 | 8.6 | 15.7 | 14.3 |
| conj | 1.6 | 2.2 | 2.5 | 1.7 |
| cop | 73.4 | 76.0 | 77.2 | 73.4 |
| csubj | 3.7 | 0.0 | 0.0 | 3.7 |
| csubj:pass | 0.0 | 0.0 | 0.0 | 0.0 |
| det | 94.4 | 95.6 | 94.8 | 94.8 |
| det:predet | 77.8 | 88.9 | 88.9 | 77.8 |
| discourse | 0.0 | 0.0 | 0.0 | 0.0 |
| dislocated | 0.0 | 0.0 | 0.0 | 0.0 |
| expl | 58.1 | 58.1 | 54.8 | 62.9 |
| flat | 8.3 | 6.1 | 7.8 | 7.4 |
| goeswith | 0.0 | 0.0 | 0.0 | 0.0 |
| iobj | 50.0 | 60.0 | 50.0 | 70.0 |
| mark | 65.6 | 67.0 | 66.7 | 65.6 |
| nmod | 7.3 | 7.8 | 8.7 | 8.5 |
| nmod:npmod | 15.8 | 15.8 | 15.8 | 15.8 |
| nmod:poss | 68.2 | 71.5 | 70.4 | 69.9 |
| nmod:tmod | 12.8 | 12.8 | 7.7 | 10.3 |
| nsubj | 54.3 | 55.8 | 58.8 | 58.5 |
| nsubj:pass | 40.2 | 45.6 | 46.9 | 41.0 |
| nummod | 72.1 | 72.8 | 73.2 | 74.0 |
| obl | 2.6 | 2.4 | 3.4 | 3.6 |
| obl:npmod | 65.0 | 70.0 | 65.0 | 70.0 |
| obl:tmod | 11.1 | 11.1 | 11.1 | 11.1 |
| orphan | 0.0 | 0.0 | 14.3 | 14.3 |
| parataxis | 0.0 | 1.0 | 0.0 | 0.0 |
| punct | 17.7 | 17.7 | 17.9 | 16.5 |
| reparandum | 0.0 | 0.0 | 0.0 | 0.0 |
| root | 100.0 | 100.0 | 100.0 | 100.0 |
| vocative | 0.0 | 0.0 | 0.0 | 0.0 |
| xcomp | 9.6 | 10.3 | 10.0 | 12.9 |
| UUAS | 52.8 | 53.7 | 54.5 | 54.0 |

Table 15: Experiment 3: Per relation results for induced unlabeled trees, comparing target only ($T.$) with increasing SSUD.

|  | Recall (EN-PUD) | | |
| UD Relation | *T.* | $k = 1$ | $k = 3$ | $k = 5$ |
|---|---|---|---|---|
| amod | 41.6 | 47.2 | 69.2 | 53.0 |
| aux | 71.0 | 64.6 | 55.1 | 72.7 |
| case | 31.0 | 41.9 | 68.1 | 56.0 |
| cc | 2.6 | 2.3 | 1.2 | 0.5 |
| compound | 50.7 | 31.0 | 15.1 | 41.2 |
| conj | 1.3 | 2.1 | 2.4 | 1.6 |
| det | 55.1 | 73.9 | 38.6 | 7.9 |
| mark | 0.5 | 0.4 | 4.5 | 0.5 |
| nmod | 3.5 | 3.3 | 2.3 | 1.6 |
| nsubj | 19.5 | 22.8 | 24.3 | 23.0 |
| nummod | 0.0 | 3.5 | 2.4 | 5.5 |
| obj | 25.0 | 27.2 | 35.6 | 36.6 |
| root | 100.0 | 100.0 | 100.0 | 100.0 |
| advmod | 5.5 | 10.2 | 8.2 | 9.4 |
| LAS | 22.48 | 25.6 | 26.3 | 22.0 |

Table 16: Experiment 3: Per relation results for induced labeled trees, comparing target only (*T.*) with increasing SSUD. Note: only labels that are considered during head selection can be labeled in the final tree.