# OpenReview forum: "Syntactic Substitutability as Unsupervised Dependency Syntax"
_EMNLP/2023/Conference — EMNLP 2023 Main_

### Official Review · Reviewer_z2Sw · 2023-08-01

**Soundness:** 4

**Excitement:**

4: Strong: This paper deepens the understanding of some phenomenon or lowers the barriers to an existing research direction.

**Paper Topic And Main Contributions:**

This work proposes a technique to boost syntactic signals to enable dependency relations to be recovered directly from an LM's attention distributions. The authors propose to mask individual words in a target sentence, and to infill these masks with a model such as BERT. The infilled sentences should tend to have isomorphic dependency structures but different vocabulary. The authors propose that averaging the attention distributions from several such infilled sentences will tend to smooth over parts of the distribution which represent non-syntactic information, while leaving shared syntactic features intact.

The authors empirically demonstrate that this averaging procedure improves both labeled and unlabeled attachment scores and increases head selection accuracy when the attention distributions are used as input features to induce a dependency parse, relative to a baseline which only uses the original, un-averaged attention distribution. Although their results are not state-of-the-art, the technique is simple, theory-agnostic, and requires no supervision.

**Questions For The Authors:**

A. On the one hand, SSUD seems to require a lot of extra passes over each sentence to create substitutions; on the other hand, these passes happen at inference time and so each one should be relatively fast. What is the overall cost of using SSUD in terms of added time?

B. Section 7.2 argues that performance on closed classes suffers when k increases, while performance on open classes increases. Is it viable to let k vary, so that larger k is used for some words and smaller k for others? Could an agent be trained to predict the optimal k for each token?

**Reasons To Accept:**

The proposed technique is simple, theory-agnostic, and requires no supervision, yet is clearly effective at distilling syntactic information from attention distributions. The general idea of using a masked LM to generate syntactically isomorphic sentences seems very general-purpose, and could probably be used for e.g. data augmentation even in supervised settings. Although the authors focus on attention distributions, in principle this technique seems suitable for distilling syntactic information from any kind of feature vector, e.g. an LM's hidden states. Theoretically, the work depends only on the intuition that instances of the same syntactic category can be substituted for one another, so it is not tied to a particular formalism or set of categories. The current work only considers individual words, but there is a clear avenue for future work which would investigate longer substitutions (potentially even including discontinuous constituents, which should be easily handled given the use of a masked LM). Overall the ideas presented in this work seem like they could be generally useful even though the work itself does not achieve state-of-the-art results.

**Reasons To Reject:**

I do not see any strong reasons to reject.

Pre-rebuttal comments:
If I have understood the proposed method correctly, it is necessary to perform a full pass over the entire sentence each time a word is substituted, and each word in the sentence will get substituted multiple times over. This seems like a disproportionate amount of extra computation relative to the observed performance gains. This issue is compounded by the fact that using too few substitutions produces *worse* results than the baseline in some settings (Table 1, albeit by a very small margin).

**Reproducibility:**

4: Could mostly reproduce the results, but there may be some variation because of sample variance or minor variations in their interpretation of the protocol or method.

**Reviewer Confidence:**

3: Pretty sure, but there's a chance I missed something. Although I have a good feel for this area in general, I did not carefully check the paper's details, e.g., the math, experimental design, or novelty.

---

> ### Author Rebuttal · Authors · 2023-08-28
>
> Dear Reviewer,
>
> We would like to thank you for your detailed review of our paper and for highlighting some of the following strengths:
> * The general utility of our methodology which is not ‘tied to a particular formalism or set of categories’
> * Future applications including: ‘distilling syntactic information from any kind of feature vector,’ ‘potentially [...] discontinuous constituents,’ and ‘data augmentation even in supervised settings.’
>
> Strengths mentioned by other reviewers include:
> * Our ‘[great] framing and formalism provided for the work,’ reviewer cSPA
> * The ‘well-written and well-structured’ paper and ‘detailed examples and error analysis,’ reviewer cSPA
> * A ‘well-founded linguistic motivation,’ reviewer 5gCw
>
> We also engage with your concerns, questions, and suggestions below:
>
> > What is the overall cost of using SSUD in terms of added time?
>
> On EN-PUD, the average wall-time to parse 1 sentence with SSUD (k=10) is 1130ms, compared to 470ms without using any substitutions. In our current implementation, we batch the substitutions at each word when performing inference. One clear way to scale this up is to perform batched inference on all substitutions of a given sentence, which we can expect to decrease the time difference even further, making our technique more viable.
>
> > Is it viable to let k vary, so that larger k is used for some words and smaller k for others? Could an agent be trained to predict the optimal k for each token?
>
> Thank you for this interesting suggestion! A possibility would be to simply vary the number of substitutions based on syntactic category (e.g. use higher k for nouns and lower for determiners) and tuning these hyperparameters using the development set. This would maintain our unsupervised setup, and the results from Experiment 3 suggest that this may be fruitful, as the reviewer has already noted. Extending SSUD to a supervised setup, it is plausible that we would get greater improvements by training a model to choose k based on more fine-grained syntactic classes. For example, transitive verbs like ‘see’ inherently form a larger class than verbs like ‘believe’. We will mention this as an avenue for future work.
>
> > Using too few substitutions produces worse results than the baseline in some settings (Table 1, albeit by a very small margin).
>
> Table 1 reports layerwise results, with SSUD performing slightly worse than the baseline (-0.1) in two instances. Previous work has shown that linguistic information, like syntax, is not uniformly present in all layers (Tenney et al, 2019). Kim et al. (2020) find that higher layers perform best for syntactic parsing, similar to what we have used. We suggest that in layers where syntactic information is not robustly represented, using substitutions may lead to random effects such as no longer predicting edges that were previously predicted due to non-syntactic information, leading to the score decrease. This is further supported by the fact that lower layers perform much worse when only non-adjacent edges are considered, e.g. Layer 7 has only 6.7 recall, compared to 24.9 for Layer 10. Given this, we believe that the **slightly worse results may say more about the model than the effectiveness of our methodology**. We will add these additional analyses within Appendix B.
>
> Based on comments from other reviewers, in the camera-ready version we will also be including results from experiments using bert-large where we find the improvements even larger, new experimental results in support of our examples and error analysis by comparing UD annotation to Surface-syntactic UD (Gerdes et al., 2018), and further discussion about sentence generation (see response to reviewer 5gCw). We will also include clarifications with respect to methodology and dataset usage (see response to reviewer cSPA).
>
> We hope that we have addressed your questions and concerns above and will add discussion about your suggestions for future work to the camera-ready version. Given this discussion, would you be willing to reconsider some of the reasons provided for rejection in your review? We are happy to engage further.

---

### Official Review · Reviewer_5gCw · 2023-08-04

**Soundness:** 4

**Excitement:**

4: Strong: This paper deepens the understanding of some phenomenon or lowers the barriers to an existing research direction.

**Paper Topic And Main Contributions:**

The paper explores a novel approach of inducing syntactic dependencies from the attention heads of Transformer-based language models. The core of the approach is to use not only the attention of the target sentence but also to average accross the attention distributions of additionally generated sentence with the same syntactic properties. The authors employ the phenomenon of syntactic substitutability to generate syntactically invariable alternative sentences. The assumption is that syntactic relations operate on word categories rather than individual words, i.e. a syntactic relation should remain the same, if the two words it connects are substituted with words from the same categories.

The authors show that their method induces slightly better dependency structures than the case when only the target sentence is considered. The authors also zoom in on a specific phenomenon, namely long-distance subject-verb relations to evaluate the effect on more difficult syntactic dependencies. By replacing the attention matrices of an existing algorithm with matrices generated via the proposed method, the authors can directly compare the results and show improvement when their method is used.

**Reasons To Accept:**

- Simple and straightforward extension of existing techniques
- Well-founded linguistic motivation of the proposed method
- A focused contribution to understanding the ways Transformer-based models capture syntactic information

**Reasons To Reject:**

Despite some merits, I tend towards rejecting the paper.

One reason is that I am not convinced that the authors indeed elevate the problem of previous methods that the relations modelled by attention are indeed syntactic in nature.  Although they had 8 pages, the authors do not elaborate on a core component of their method, namely the sentence generation procedure. They mention BERT and the usage of POS tagger to control the generated sentences but this may not be enough. For example, BERT can easily generate synonyms or very closely related words, e.g., love-like or thought-figured. In such cases, the attention will probably model the close semantic relation rather than a syntactic one.

Another reason is the lack of generalisation accross models. The average attention mechanism the authors propose can easily be tried with  other models. After all, we cannot draw reliable conclusions about how LLMs model syntactic information by the random fact that in this experiment setup it seemed that syntactic information was sufficiently captured in BERT Layer 10. Without more details provided or even assumptions, it seemed to me that the results are quite random and do not really contribute to understanding how LLMs model syntactic information.

In general, I think that the paper would have had better chances as a short paper submission. The contribution and the experiments are rather small and focused. The authors spend a lot of space explaining obvious things or terms as well as repeating information, especially with regard to previous work. Instead, the paper could have easily been fit into 4 pages, with more details on the core parts of the approach.

**Reproducibility:**

5: Could easily reproduce the results.

**Reviewer Confidence:**

5: Positive that my evaluation is correct. I read the paper very carefully and I am very familiar with related work.

---

> ### Author Rebuttal · Authors · 2023-08-28
>
> Dear Reviewer,
>
> We would like to thank you for your detailed review of our paper and highlighting strengths like:
> * ‘Well-founded linguistic motivation,’ also reviewer cSPA
> * ‘Simple and straightforward extension of existing techniques,’ also reviewer z2Sw
>
> Strengths other reviewers mentioned include:
> * ‘[Great] framing and formalism,’ ‘well-structured’ paper and ‘detailed examples and error analysis,’ reviewer cSPA
> * ‘General-purpose’ nature  ‘e.g. data augmentation even in supervised settings,’ reviewer z2Sw
>
> We discuss the criticism of the implementation and motivation:
> > do not elaborate on a core component… BERT can easily generate synonyms… e.g., love-like or thought-figured… attention will probably model the close semantic relation
>
> We respectfully disagree. We test whether substitution enables better extraction of syntax, drawing on the intuition that syntactic information is shared across different substitutions. BERT can generate synonyms, however, we hypothesize and show that more substitutions obviates the effect: in Tab. 2, 4, & 5, parsing scores increase with more substitutions. We do not claim that single substitutions effectively instantiate syntactic classes for parsing.
>
> We point out that **‘thought-figured’ and ‘like-love’ are closely related due to syntactic constraints**. ‘Thought’ can only be substituted by verbs embedding full clauses, which often express speaker stance. No verbs like ‘eat’ embed full clauses. Substituting verbs that take different complements leads to structure change or ungrammaticality. **As desired, ‘thought-figured’ and ‘like-love’ reflect fine-grained syntactic categories.** Words from larger syntactic categories contrast this, e.g. 'know-help' (Fig. 3).
>
> In the camera-ready version, we will address these concerns by
> * providing more examples of generated substitutions in Fig. 3,
> * elaborating on proper substitutions, discussing considerations like verb subcategorization.
>
> > the random fact that... syntactic information was sufficiently captured in BERT Layer 10... do not really contribute to understanding how LLMs model syntactic information.
>
> We would like to point to our stated motivation (80-84) that we investigate **substitutability for extracting syntactic information in parsing**. We argue that modelling substitutability better captures how syntax is ontologically defined. It follows that unsupervised induction enables future work to compare model-derived structures and human annotations.
>
> Our choice to use BERT-base stems from the vast literature using it for extracting unsupervised syntax (Kim et al. 2020, Limisiewicz et al. 2020, Zhang and Hashimoto 2021) and probing (Hewitt and Manning 2019, Clark et al. 2020). Following a suggestion from the reviewer, we also tested another model BERT-large and will report results from **experiments in the camera-ready version**. In BERT-large, we find **SSUD improves parsing by 1.4pts (Layer 17) and 1.9pts (Layer 18)**. Layer 17 UUAS is 46.9.
>
> **We also strongly contend that these results are not ‘random.’** Instead, our finding that dependency information is centered at Layer 10 of BERT-base and Layer 17/18 of BERT-large **corroborates previous work from Kim et al (2020)** who find constituency information is centered at Layer 9 (base) and Layer 16 (large). Since Tenney et al (2019) show that constituency occurs before dependency, our findings are well-grounded in the existing literature.
>
> To address these concerns, we will improve our error analysis of induced parses (Sec 5.3, Fig.4 and App. B) where **we observe a tendency towards relations headed by function words** with following experiment: induced parses on EN-PUD are scored using two annotations, UD and Surface-Syntactic Universal Dependencies (SUD, Gerdes et al., 2018). Unlike in UD, **SUD uses function words as heads**. Here, we find strong experimental support: **scored on SUD, UUAS is 12.6pts higher than on UD (59.0 vs 46.4).** SSUD provides a 3.0pt improvement over the baseline when assessed on SUD, compared to the 2.1pt improvement with UD.
>
> Given this, we find that SSUD provides solid improvements, not just ‘slightly better.’ Improvements of 3.0pts in unsupervised parsing are on par with previous work (e.g. +2.8pts over baseline; Zhang and Hashimoto, 2021). Relation level results (Experiment 2 and 3) and the experiment above show how annotation differences can obfuscate improvements.
> > paper could have easily been fit into 4 pages
>
> In order to cover the theoretical motivation and experiments on multiple datasets, this would require a longer appendix. We find this less desirable.
>
> In summary, we will (i) further discuss substitution generation, and (ii) provide new experiments to evaluate SSUD on BERT-large and give further evidence that SSUD has systematic improvements for parsing in the camera-ready version. We hope we have addressed your concerns point-by-point.
>
> In light of the clarification and these improvements, would you reconsider the evaluation of our work and raise the scores? We are happy to engage further.
>
> **Additional references:**
>
> Kim Gerdes, Bruno Guillaume, Sylvain Kahane, and Guy Perrier. 2018. SUD or Surface-Syntactic Universal Dependencies: An annotation scheme near-isomorphic to UD. In Proceedings of the Second Workshop on Universal Dependencies (UDW 2018), pages 66–74, Brussels, Belgium. Association for Computational Linguistics.

---

### Official Review · Reviewer_cSPA · 2023-08-05

**Soundness:** 4

**Excitement:**

4: Strong: This paper deepens the understanding of some phenomenon or lowers the barriers to an existing research direction.

**Paper Topic And Main Contributions:**

This paper introduces a new method for unsupervised dependency parsing, based on syntactic substitutability (using the ability to substitute a given word as a means to consolidate the estimation of corresponding subtrees). It presents both a theoretical framework for grounding the method, and experiments to evaluate it empirically. Experiments show promising results for the method in general, and huge improvements on harder constructions of long-distance agreement. The paper also includes first positive results on combining that method with other existing methods for unsupervised dependency parsing.

**Questions For The Authors:**

A - Equation (5): Since the dependencies are undirected, why are the substitutions done only on i, and not j as well?

B - Line 369: is there a motivation in particular to use a dataset with so short sentences? Is it more informative, easier to interpret, more representative of the targeted phenomena...? Since Experiment 2 focuses on long-distance agreement, the use of WSJ10 here is surprising.

C - Experiment 1.1 selects layer 10, but which head (cf. multi-head attention)? How is the head selected? Or are the attentions also aggregated over heads of the same layer?

**Reasons To Accept:**

The proposed methodology is a very interesting idea that sheds new light on unsupervised parsing. It also benefits from solid theoretical foundations, and the experiments are organized, conducted and analyzed with a rigorous methodology.

The problem statement is carefully laid out in §3.1, and more generally the framing and formalism provided for the work are great.

The detailed analysis made of the results of Experiment 3 (lines 578 and beyond) is another token of that good methodology. I appreciated the detailed examples and error analysis in Annex C as well.

The paper itself is well-written and well-structured.

Annex E denotes extensive efforts for reproducibility (documenting as far as the package versions). And the (too rare) discussion on the licensing of the data is worth noting (annex F).

**Reasons To Reject:**

~~Beyond minor presentation issues, the only shortcoming is a lack of clarity of the methodology regarding the step of head selection (see Question C below).~~ Only minor presentation issues

**Reproducibility:**

4: Could mostly reproduce the results, but there may be some variation because of sample variance or minor variations in their interpretation of the protocol or method.

**Reviewer Confidence:**

4: Quite sure. I tried to check the important points carefully. It's unlikely, though conceivable, that I missed something that should affect my ratings.

**Typos Grammar Style And Presentation Improvements:**

- The mention of "increasing the number of substitutions" in the abstract (line 17) is confusing, because at that stage the reader does not know that k=1 in Experiment 1.1. Either rephrase to better contextualize the idea, or else avoid mentioning in the abstract this point about the increase.
- There are inconsistencies in the numbering of Figures and Tables (there is Figure 1 vs Table 1, but then suddenly there is no Table 4 because there is Figure 4).
- Line 210, "all words in **the** sentence" would read better (= the same sentence as the one in line 209)
- Typo line 234: substitability --> substitutability
- Typo line 396: number --> number of
- Line 426, "As predicted, this may reflect" is awkward. Do those results confirm an hypothesis, and which one? Or is this about restating the hypothesis made (on how to interpret such numbers)? It should read either as "As predicted, this reflects", or "As discussed, this may reflect"
- Typo line 550: direction --> directions
- Typo line 851: remove "of"
- Typo line 854: remove "is"
- Typo line 856: remove "that the"?

---

> ### Author Rebuttal · Authors · 2023-08-28
>
> Dear Reviewer,
>
> We would like to first thank you for your detailed review of our paper and are grateful for your suggestions to improve the presentation which we will fix for the camera-ready version. Thank you as well for highlighting strengths such as:
> * ‘Solid theoretical foundations,’ also noted by reviewer 5gcw
> * A ‘carefully laid out’ problem statement and ‘[great] framing and formalism’
> * The ‘well-written and well-structured’ paper
>
> We would like to mention some strengths observed by other reviewers, including:
> * Its ‘general-purpose’ nature  ‘e.g. data augmentation even in supervised settings,’  and ‘clear avenue for future work’ reviewer z2Sw
> * ‘Simple and straightforward idea,’ reviewers 5gcw and z2SW
>
> We appreciate your questions that will serve to clarify our paper’s methodology further. We address them below:
>
> >A: Equation (5): Since the dependencies are undirected, why are the substitutions done only on i, and not j as well?
>
> Eq. (5) makes reference to the attention distribution of a sentence, where only 1 word is substituted at a time in order to better control for possible changes in syntactic structure. As you have correctly stated, dependencies are undirected, so the choice of i or j in this equation is arbitrary. In fact, since the final matrix aggregates substitutions at each position when parsing, the MST algorithm does use the attention scores resulting from substituting both i and j of any given syntactic relation. We will add discussion to this effect to clarify this equation. This question also sparks an interesting possibility for future work, where substitutions are done in a pairwise fashion throughout the sentence and a more complex selection of attention distributions is performed. This idea would seem to converge towards the mutual information approach explored in Hoover et al. (2021).
>
> >B: Is there a motivation in particular to use a dataset with so short sentences? Is it more informative, easier to interpret, more representative of the targeted phenomena...? Since Experiment 2 focuses on long-distance agreement, the use of WSJ10 here is surprising.
>
> The WSJ10 dataset is used in Experiment 1 in order to compare against baselines from previous work which also test this exact dataset. But for the long-distance agreement (Experiment 2), we use sentences from Marvin and Linzen’s (2018) long-distance agreement dataset, rather than the shorter sentences from WSJ10. We will clarify further in Section 4.
>
> >C: Experiment 1.1 selects layer 10, but which head (cf. multi-head attention)? How is the head selected? Or are the attentions also aggregated over heads of the same layer?
>
> The attention distributions are aggregated over heads of the same layer (389-390). This choice follows work from Kim et al. (2020) who do similarly in the context of syntactic constituency parsing (433-434). We will clarify this methodological point in Sec 5.1 (Experiment 1) and further discuss how the layer we use corroborates previous findings in the aforementioned paper. Experiment 3 complements the use of layerwise attention distributions in Experiment 1 and explores SSUD with a method that aggregates scores from heads chosen across different layers (Limisiewicz et al, 2020).
>
> Additionally, thank you for raising this suggestion about the abstract:
> > mention of "increasing the number of substitutions" in the abstract (line 17) is confusing
>
> We will rephrase the abstract in order to make it clear that substitutions are used to model syntactic substitutability before referring to increasing the number of substitutions.
>
> Based on comments from reviewer 5gCw, we will also be including results from experiments using an additional model (BERT-large) to further support our hypothesis, new experimental results in support of our examples and error analysis by comparing UD annotation with Surface-Syntactic UD (Gerdes et al., 2018), and further discussion about sentence generation in the camera-ready version. In summary, we find strong trends with BERT-large, and even larger improvements when we test on Surface-Syntactic UD.
>
> We hope to have sufficiently answered each of your questions and will add clarification with respect to the datasets and attention head selection to the camera-ready paper as well. Given your assessment and these improvements, would you consider championing our paper for publication at this conference? We are happy to engage further.

---

### Meta-Review · Area_Chair_AdFU · 2023-09-12

**Recommendation:** 5

**Metareview:**

This paper presents a novel unsupervised dependency parsing approach based on LLM's attention heads. It offers a theoretical foundation for the method and empirical evaluations, revealing especially promising results for more challenging phenomena, such as long-distance subject-verb agreement.

In the case of this submission, the scores of the reviewers are unanimous. It is a **strong submission** -- both regarding soundness (score 4) and excitement (score 4). Among the strengths of the paper, the reviewers mentioned a lot of very important points:
* solid theoretical foundations
* simple and well-targeted method
* great methodology, well-written and well-structured paper
* detailed analysis helping to understand a particular issue
* clear future work directions

After the first round of reviews, the biggest weakness of the paper were unclear details (e.g., sentence generation, head selection). However, as reviewer 5gCw said in the discussion, *"the authors went above and beyond in their response to answer [reviewer's] criticism"*. As a result, the concerns of two reviewers eased after the rebuttal phase. They encourage the authors to clarify the details of the paper and improve smaller presentation issues for the camera-ready version.

---

### Decision · Program_Chairs · 2023-10-07

**Decision:**

Accept-Main

**Comment:**

This paper presents a novel unsupervised dependency parsing approach based on LLM's attention heads. It offers a theoretical foundation for the method and empirical evaluations, revealing especially promising results for more challenging phenomena, such as long-distance subject-verb agreement.

In the case of this submission, the scores of the reviewers are unanimous. It is a **strong submission** -- both regarding soundness (score 4) and excitement (score 4). Among the strengths of the paper, the reviewers mentioned a lot of very important points:
* solid theoretical foundations
* simple and well-targeted method
* great methodology, well-written and well-structured paper
* detailed analysis helping to understand a particular issue
* clear future work directions

After the first round of reviews, the biggest weakness of the paper were unclear details (e.g., sentence generation, head selection). However, as reviewer 5gCw said in the discussion, *"the authors went above and beyond in their response to answer [reviewer's] criticism"*. As a result, the concerns of two reviewers eased after the rebuttal phase. They encourage the authors to clarify the details of the paper and improve smaller presentation issues for the camera-ready version.